# Transcriptome Analysis to Elucidate the Effects of Milk Replacer Feeding Level on Intestinal Function and Development of Early Lambs

**DOI:** 10.3390/ani13111733

**Published:** 2023-05-24

**Authors:** Guoxiu Wang, Qian Zhang, Zhanyu Chen, Yongliang Huang, Weimin Wang, Xiaoxue Zhang, Jiale Jia, Qihao Gao, Haoyu Xu, Chong Li

**Affiliations:** 1College of Animal Science and Technology, Gansu Agricultural University, Lanzhou 730070, China; wanggx@gsau.edu.cn (G.W.); chenzy_gsau@163.com (Z.C.); huangyl_gau@163.com (Y.H.); zhangxx@gsau.edu.cn (X.Z.); jiajiale0927@163.com (J.J.); gsau_gqh@126.com (Q.G.); xuhy_gsau@163.com (H.X.); 2Institute of Grassland Research of Chinese Academy of Agricultural Sciences, Hohhot 010000, China; zhangqian05@caas.cn; 3State Key Laboratory of Grassland Agro-Ecosystems, College of Pastoral Agriculture Science and Technology, Lanzhou University, Lanzhou 730000, China; wangwm@gsau.edu.cn

**Keywords:** lambs, intestine, growth performance, transcriptome sequencing

## Abstract

**Simple Summary:**

Intestinal development is a gradual process that starts before birth and continues throughout the postnatal period, with nutrition playing a critical role. In this study, we performed transcriptome sequencing to investigate the effects of milk replacer (MR) feeding level on intestinal gene expression in lambs and its relationship with intestinal function and development. The intensive use of MR was observed to promote intestinal morphological development and digestive enzyme activities and considerably affect intestinal gene expression, which was still significant at 14 days postweaning. Further, the intensive use of MR affected the insulin sensitivity of the intestinal tissue and regulated nutrient distribution and metabolism by synchronizing the expression of *AHSG*, *IGFBP1*, *MGAT2*, *ITIH*, and *CYP2E1* in the jejunal tissue of lambs.

**Abstract:**

Although early feeding strategies influence intestinal development, the effects of milk replacer (MR) feeding level on intestinal structure and functional development and underlying regulatory mechanisms remain unclear. In this study, 14 male Hu lambs were fed MR at 2% or 4% of their average body weight and weaned at 35 days of age. The MR was produced by the Institute of Feed Research of the Chinese Academy of Agricultural Sciences, and it contains 96.91% dry matter, 23.22% protein, and 13.20% fat. Jejunal tissues were assessed by RNA-seq for differences in the gene expression of lambs at 49 days of age; regulatory pathways and mechanisms of the effects of early nutrition on intestinal function and development were analyzed, along with growth performance, feed intake, jejunal histomorphology, and digestive enzyme activities. Increasing MR- feeding levels increased dry matter intake and daily gain before weaning, as well as lactase, amylase, lipase, trypsin, and chymotrypsin activities and intestinal villus length and muscular thickness. Overall, 1179 differentially expressed genes were identified, which were enriched in nutrient metabolism, coagulation cascades, and other pathways. Further, intensive MR feeding affected insulin sensitivity to reduce excessive glucose interception by intestinal tissues to ensure adequate absorbed glucose release into the portal circulation and promoted lipid and protein degradation in intestinal tissues to meet the energy demand of intestinal cells by regulating *AHSG*, *IGFBP1*, *MGAT2*, *ITIH*, and *CYP2E1* expression.

## 1. Introduction

Milk replacers (MRs) are widely used in lamb rearing and are formulated to meet the specific nutritional requirements of lambs [1,2,3]. They are easy to prepare and feed and can be successfully used to rear orphaned or rejected lambs, as well as supplement the milk production of ewes with multiple lambs or those with insufficient milk secretion [2]. The rational utilization of MRs is to improve the survival rate and growth performance of lambs. Malnutrition during early life, including undernutrition or overnutrition, can cause a range of adverse health outcomes for lambs [4]. Suitable nutrient and feeding levels are the key to the rational application of MRs in lamb production. In lamb raising, MRs are usually quantitatively fed, but their feeding level varies considerably from farm to farm; thus, the feeding level of MRs is more important to the growth and development of lambs than the nutrient level. 

The intensive use of MRs can reportedly promote lamb weight gain and improve immunity; however, at the same time, it can reduce the starter intake, which is not conducive to rumen development [5,6]. Furthermore, nutrients required for optimal growth performance in young animals may not be the same as those required for optimal intestinal development and healthy immune system maintenance [7]. The intestine is crucial for nutrient digestion and absorption and for protection against external pathogens [8]. Therefore, understanding feeding and management factors that affect intestinal development during the first few weeks of life is important to improve animal health, welfare, and performance [9,10]. Intestinal morphological development is critical for the health, growth, and development of young animals. The greater morphological development of villi and an increased surface area is important for nutrient absorption to support increased weight gain and feed-conversion efficiency [11]. Sauter et al. evaluated the effects of glucocorticoids and colostrum supply on intestinal morphology in neonatal calves and found it to be responsive to milk feeding strategy or colostrum amount and quality [12]. Further, Ontsouka et al. found that major morphological and functional changes in the calf intestine are initiated by colostrum bioactive substances [13].

Although these studies suggest that early feeding strategies influence intestinal morphological development, the effects of the MR feeding level on intestinal structure and functional development and underlying regulatory mechanisms remain unclear. Intestinal development is a gradual process that starts before birth and continues throughout the postnatal period, with nutrition playing a critical role [14]. Intestinal epithelial cells are the most rapidly renewing and metabolizing cells in the body [15]. The absorption of nutrients by intestinal epithelial cells and the establishment of an intestinal barrier entail a remarkably high demand for both substance and energy [16]. Nutrient intake evidently has vital effects on intestinal cell metabolism and proliferation. Therefore, we hypothesized that the MR feeding level affects the intestinal development of lambs through transcriptional regulation and has a lasting effect. We analyzed the effects of the MR feeding level on growth performance and feed intake of lambs pre- and postweaning; moreover, we evaluated intestinal development and functionality by assessing intestinal morphology and digestive enzyme activities. Furthermore, transcriptome sequencing was performed to investigate the effects of the MR feeding level on intestinal gene expression and its relationship with intestinal function and development. We believe that a detailed understanding of the regulation of intestinal development could provide the basis for the rational use of early nutritional regulation strategies to improve the productivity and health of lambs.

## 2. Materials and Methods

### 2.1. Experimental Design and Animal Management

Fourteen male Hu lambs with similar birth weight (3.29 ± 0.68 kg) were selected; each lamb was born to ewes with a litter size of 2. These lambs were randomly divided into two groups. The control group (group C, *n* = 7) was fed a standard quantity of MR, which was 2% of their average body weight (BW) per day (Beijing Precision Animal Nutrition Research Center, Beijing, China) [17], while the intensive MR feeding group (group H, *n* = 7) was fed an increased quantity of MR, which was 4% of their BW per day. From birth to day 3, the lambs were kept indoors with the ewes to ensure adequate colostrum intake. From day 4 to 6, the lambs were trained to use the nippled bottle containing reconstructed MR (23.22% crude protein and 13.20% fat, air-dried basis; produced by the Institute of Feed Research of the Chinese Academy of Agricultural Sciences, Beijing, China; Table 1), while still being raised with the ewes and receiving breast milk. The MR was reconstituted at 200 g/L in water and provided to lambs at 40 °C. Each lamb was fed 50 mL MR thrice a day (at 09:00, 15:00, and 21:00). At 7 days of age, the lambs were separated from the ewes and housed in individual pens (0.65 m × 1.10 m); lactation was completely replaced by artificial feeding with MR. According to the experimental design, the MR was artificially fed as per the corresponding feeding amount. The starter diet was added at 7 days of age. At day 35, all lambs were weaned from MR and fed only the starter diet, which was formulated according to the Feeding Standard of Meat-Producing Sheep and Goats (NYT816-2004) published in China. Table 1 shows the formula and nutritional composition. All lambs had free access to the starter diet and clean water.

### 2.2. Measurement of Growth Performance and Feed Intake

All lambs were weighed at birth and then weighed every 7 days to adjust MR feeding scale. Starting from the age of 7 days, the starter intake of each lamb was recorded daily as the difference between the offered and refused feed. The BW, average daily gain (ADG), average daily dry matter (DM) intake, and feed conversion ratio (FCR, daily DM intake/ADG) of lambs before (from 7 to 35 days of age) and after (from 35 to 49 days of age) weaning were calculated.

### 2.3. Sample Collection

All experimental lambs were slaughtered at 49 days of age after a 12-h fast. Middle jejunum samples were immediately collected post-slaughter. After rinsing with PBS, one jejunum sample was fixed in 4% paraformaldehyde for histomorphological analysis, and another jejunum sample was collected in a sterile freezer tube, rapidly frozen in liquid nitrogen, and stored at −80 °C for total RNA extraction. We selected the jejunum as the target tissue for RNA-seq analysis due to its relatively greater length in comparison to other intestinal segments in sheep and its crucial role in facilitating the absorption of nutrients. Jejunal contents were collected in sterile tubes and stored at −80 °C for enzymatic activity analysis. 

### 2.4. mRNA Library Construction and Sequencing

RNA was extracted from the intestinal tissue samples with the MiniBEST Universal RNA Extraction kit (TaKaRa, Kusatsu, Japan), according to manufacturer instructions. The RNA concentration was measured using a NanoDrop 2000 spectrophotometer (Thermo Scientific, Wilmington, NC, USA), and the RNA integrity was measured by 1% agarose gel electrophoresis. The quality and purity of the total RNA was further tested using Bioanalyzer 2100 and a LabChip kit (Agilent, Santa Clara, CA, USA), ensuring that all RNA samples had RIN values of >7.0. Approximately 10 ug of total RNA was subjected to isolate Poly (A) mRNA with poly-T-oligo attached magnetic beads (Invitrogen, Carlsbad, CA, USA). Following purification, the mRNA was fragmented into small pieces utilizing divalent cations at an elevated temperature. These fragments were then reverse-transcribed, and the final cDNA library was obtained by following the protocol for the mRNA Seq sample preparation kit (Illumina, San Diego, CA, USA); the average insert size for paired-end libraries was 300 bp (±50 bp). Subsequently, paired-end sequencing was performed on Illumina HiSeq 4000, according to manufacturer instructions.

Raw data generated by sequencing was filtered with Cutadapt (v1.4) [18] to exclude unqualified sequences. Adapter reads, reads with undetermined base information in the total number of raw reads exceeding 5%, and low-quality reads (base number of mass value Q ≤ 10 accounting for >20% of the whole read) were removed. Clean reads were then obtained by verifying the sequence quality using FastQC (v0.10.1), including Q20, Q30 and GC content. All downstream analysis was performed using high-quality clean data. Raw sequence data have been submitted to the NCBI Short Read Archive, with accession number PRJNA938493.

### 2.5. Reference Genome Alignment and Differential Gene Expression Analysis

We aligned reads to the UCSC (http://genome.ucsc.edu/, accessed on 10 March 2022) sheep reference genome by means of the HISAT package, which initially removes a portion of reads based on their quality and then maps them to the reference genome. HISAT builds a database of potential splice junctions and confirms them by comparing previously unmapped reads against the database of putative junctions. The mapped reads of each sample were assembled using StringTie (v1.3.0) [19], and all sample transcriptomes were then merged to reconstruct a comprehensive transcriptome using perl scripts. StringTie was employed to assess the mRNA expression level by calculating FPKM [19]. Differentially expressed mRNAs and genes (DEGs) were identified by an R package based on |log_2_ (fold change (FC))| > 1 and *p* < 0.05.

### 2.6. Quantitative Reverse-Transcription PCR (qRT-PCR)

To verify the differential gene expression results, qRT-PCR was performed for nine DEGs across all RNA samples. Primer Premier 6.0 (Premier Biosoft Interpairs, Palo Alto, CA, USA) was used to design primers (Appendix A). Samples of cDNA obtained by reverse transcription were continuously diluted to determine their amplification efficiency, and primers with an amplification efficiency > 90% were used. Furthermore, β-actin served as the reference gene, and the relative expression levels of candidate genes were detected by real-time PCR (Bio-Rad Laboratories Inc., Hercules, CA, USA). Each 25 μL PCR mixture comprised 2 μL cDNA, 12.5 μL SYBR Premix Ex Taq (TaKaRa, Kusatsu, Japan), 0.4 μL forward and reverse primers each, and 9.7 μL ddH_2_O. The cycling conditions were as follows: 95 °C for 30 s, 40 cycles of 95 °C for 5 s, and 60 °C for 30 s. Three replicates were assessed for all samples to calculate the relative expression levels of candidate genes. The fold change in the gene expression was calculated as 2^−ΔΔCt^ [20].

### 2.7. Pathway Analysis of DEGs

Gene ontology (GO) analysis was performed to determine significantly enriched biological processes and groups of DEGs. Further, the Kyoto Encyclopedia of Genes and Genomes pathway database (KEGG, http://www.genome.jp/kegg/, accessed on 10 March 2022) was used to identify pathways that were the most significant to the dataset.

### 2.8. Construction of a Gene Co-Expression Network and Identification of Significant Modules

We performed weighted gene co-expression network analysis (WGCNA) to construct a gene co-expression network [21]. Gene expression data were loaded into R. The pickSoftThreshold function was employed to choose an appropriate thresholding power. The topological overlap matrix, which measures the degree of connectivity between genes and is used to identify gene modules [21], was calculated using adjacency values. Gene modules were then identified using a hierarchical clustering of topological overlap matrix. Module identification was accomplished by dynamic tree cut; the minimum module size was set to 30. Modules with high similarity scores were combined with a threshold value for each dataset. Each module represents a group of highly interconnected genes. 

### 2.9. Statistical Analysis

Data were analyzed with SPSS (v21.0; IBM Corp., Armonk, NY, USA). An independent samples *t*-test was applied to analyze differences in growth performance, feed intake, jejunal morphology, and digestive enzyme activity. Group differences were considered significant when *p* < 0.05, and tendencies were discussed when 0.05 < *p* < 0.10.

## 3. Results

### 3.1. Growth Performance and Feed Intake

Before weaning, the ADG of lambs in group H at 35 days of age was significantly higher than that of group C (*p* < 0.05), and the BW of group H was higher than that of group C (*p* < 0.1). After weaning, the ADG of group H was higher than that of group C (*p* < 0.05), but there was no significant difference in BW between the two groups at 49 days of age (*p* > 0.1). Furthermore, there was no significant difference in the starter intake between the groups either before or after weaning (*p* > 0.1). The DM intake in group H was significantly higher than that in group C before weaning (*p* < 0.05). The FCR after weaning in group H tended to be higher than that in group C (*p* < 0.1; Table 2).

### 3.2. Jejunal Histomorphology and Digestive Enzyme Activity

The villi length and muscle thickness of the jejuna in group H were significantly higher than those in group C (*p* < 0.05; Table 3, Figure 1). In addition, in group H, the activities of lactase, amylase, lipase, tryptase, and chymotrypsin in the jejuna were significantly higher than those in group C (*p* < 0.05; Table 3).

### 3.3. Sequencing Data-Mapping Statistics

In total, >655 million clean reads were obtained from 14 samples after adapter trimming and quality filtering; of these, >585 million reads were mapped to the sheep transcriptome and genome. Almost 59.27–62.32% reads aligned in a unique manner, and 25.72–30.71% classified as multiple-mapped reads. Detailed information pertaining to the sequencing data mapping statistics is presented in Appendix A.

### 3.4. Analysis of DEGs

A total of 26,247 annotated genes were defined as being expressed in the jejunum of experimental lambs. In total, 1179 DEGs (*p* < 0.05) with an FC of >2 were identified. Relative to group C, 882 DEGs were up- and 297 DEGs were downregulated in group H (Figure 2). Appendix A shows the description of DEGs, and Table 4 shows the top 20 DEGs (19 up- and 1 downregulated) with the highest FC.

To validate the differential expression of genes, nine genes were subjected to qRT-PCR. As shown in Appendix A, the expression patterns of five significantly upregulated DEGs (*AHSG*, *ACAT1*, *FADS1*, *INSIG1*, and *PCK1*) and four significantly downregulated DEGs (*MGAT2*, *CD79B*, *CXCL13*, and *MS4A1*) were consistent with sequencing data, implying that the results of transcriptome sequencing and analysis were highly reliable.

### 3.5. Enrichment Analysis of DEGs

To explore the biological functions of DEGs, GO enrichment and KEGG pathway analyses were performed. The functional enrichment analysis of GO terms for all 599 genes is shown in Figure 3A. DEGs were significantly enriched in 1397 GO entries. In terms of biological process, DEGs were mainly enriched in oxidation–reduction processes, negative regulation of endopeptidase activity, metabolic process, steroid metabolic process, fibrinolysis, one-carbon metabolic process, complement activation, cell-matrix adhesion, lipid metabolic process, among others. KEGG pathway analysis (Figure 3B) showed that the most enriched categories included complement and coagulation cascades, metabolism of xenobiotics by cytochrome P450, amino acid biosynthesis, steroid hormone biosynthesis, and metabolism of various amino acids, among others.

### 3.6. Construction of a Gene Co-Expression Network and Identification of Significant Modules

We performed WGCNA with jejunal gene expression data. Genes were clustered based on their connectivity, and five modules of genes with a co-expression pattern were identified. Figure 4 shows the color assigned to each module, plus the correlation coefficient between the eigengene values of these modules and jejunal histomorphological index and digestive enzyme activity. The gray module, which included 136 genes, was negatively correlated with jejunal villus height (r = −0.7, *p* = 0.005), muscle thickness (r = −0.69, *p* = 0.007), and lactase (r = −0.89, *p* < 0.001), amylase (r = −0.83, *p* < 0.001), lipase (r = −0.75, *p* = 0.002), trypsin (r = −0.75, *p* = 0.002), and chymotrypsin (r = −0.86, *p* < 0.001) activities. The turquoise module, which included 136 genes, was positively correlated with lactase (r = 0.55, *p* = 0.04) and chymotrypsin (r = 0.64, *p* = 0.01) activities.

## 4. Discussion

Nutrition is one of the most critical factors affecting the weight gain of young lambs. Adequate nutrition is essential for healthy growth and development, and a balanced diet with the right amount of protein, energy, minerals, and vitamins is crucial for optimal weight gain. In the current investigation, lambs in group H showed increased DM intake before weaning, and the nutrients in MR were easier to digest and absorb than those in solid feed; this was the main reason for the higher ADG of lambs in group H. The H group exhibited increased preweaning ADG compared to the C group, yet there was no significant difference in the lambs’ BW between the two treatment groups by day 49. Although there was no significant difference in DM intake after weaning, lambs in group H tended to exhibit lower ADG and higher FCR than those in group C, which may be because the lambs in group H were more susceptible to weaning stress. Studies have shown that weaning stress can be recovered in the short term, and the adverse effects do not usually last long [5,22].Moreover, research has indicated that intensive MR feeding has detrimental effects on rumen development [6], which could be a contributing factor to the temporary decline in growth performance observed in lambs after weaning. Nevertheless, we found that high MR feeding levels improved the intestinal digestive enzyme activity and tissue morphological development in lambs, and this effect was still observed 14 days postweaning. Intestinal development is vital for the growth, health, and overall well-being of young lambs. The rapid proliferation and active migration of cells are essential for a steady-state epithelial turnover in the gut and the maintenance of intestinal villus height. A healthy and well-developed intestine can efficiently absorb nutrients, leading to optimal growth and weight gain, and decrease susceptibility to diseases. A high MR feeding level can promote intestinal development and digestive function, which is of great significance for lambs.

Transcriptome sequencing was performed to further investigate the effects of the MR feeding level on the intestinal function of lambs, which led to the identification of 1179 DEGs. A large number of DEGs indicated that MR feeding levels had an important effect on intestinal transcriptional regulation. Santos et al. evaluated liver transcriptomic and proteomic profiles of preweaning lambs, and their data suggested that the early nutrient level of lambs affects nutrient distribution and utilization, leading to changes in their growth performance later in life [23]. It is notable that although the feeding patterns of lambs in the two groups were completely consistent after weaning at 35 days of age, a large number of DEGs in the intestinal tissue at 49 days of age indicated that early feeding of MR had a sustained effect on the intestinal function of lambs; this sustained effect was consistent with intestinal morphological and enzyme activity indices. As intestinal absorption and barrier function play an important role in lamb physiology, the sustained effects of early MR feeding level on intestinal function and development cannot be ignored.

Based on the results of the GO-term enrichment analysis, it was observed that in terms of biological processes, DEGs were mainly involved in the digestion, absorption, transport, and metabolism of nutrients, as well as the catalytic activities of digestive enzymes. The intestinal epithelium has a high energy demand due to the active transport of nutrients across the cell membrane, constant cell renewal and repair, and maintenance of a tight barrier between the lumen and underlying tissue [24]. This high energy demand is met by a combination of glucose, fatty acid, and amino acid metabolism [24]. Our results indicated that the MR feeding level has an important effect on intestinal nutrient metabolism. Santos et al. evaluated liver transcriptomic profiles to report that early nutrition affects pathways such as the REDOX process, amino acid metabolism, and cell apoptosis [23]. It is notable that in this study, among the top 20 KEGG pathways with the highest FC, 10 pathways were related to amino acid metabolism. Furthermore, multiple pathways involved in amino acid catabolic metabolism were significantly upregulated; jejunal trypsin and chymotrypsin activities showed an increase, and the expression levels of intestinal peptidase genes were significantly upregulated. Protein absorption and transport and amino acid biosynthesis pathways were also among the most significantly upregulated pathways. This indicated that a high MR feeding level significantly promoted amino acid metabolism in intestinal tissues. Studies have shown that only 56% of the essential amino acid (EAA) intake appeared in the portal blood, and roughly one third of the dietary intake of EAA is consumed in first-pass metabolism by the intestine [25]. Most such intercepted essential amino acids are absorbed and utilized by intestinal tissues for catabolism through transamination and decarboxylation. The catabolism of amino acids generates ATP and provides the building blocks for the synthesis of new molecules [26,27,28]. This is specifically important for the intestinal epithelium, which has a high energy demand due to its rapid turnover and constant renewal [24]. The simultaneous increase of amino acid absorption, transport, synthesis, and catabolism ultimately increases intestinal cell metabolism, which heavily relies on aerobic respiration to produce ATP. Amino acid catabolism seems to play a vital role in maintaining the physiological function of intestinal cells at high metabolic levels.

Due to the large number of DEGs identified in this study, we combined DEG enrichment analysis and WGCNA results to identify and focus on those showing the highest FC. *AHSG* (log_2_FC = 11.88), encoding alpha-2-HS-glycoprotein (aka fetuin-A), was the most significantly upregulated gene. The expression of fetuin-B (*FETUB*) was also significantly upregulated. Fetuin is a glycoprotein that plays a key role in regulating both carbohydrate and fat metabolism by leading to insulin resistance and promoting the formation of calcium phosphate crystals in adipose tissues [29]. A previous study detected significant upregulation of *AHSG* expression in diet-induced obesity rat models [29]. Some studies have suggested that elevated levels of *ASHG* can potentially lead to higher blood sugar levels caused by insulin resistance [30]. Interestingly, the genes with the second and third highest FC in this study, namely *IGFBP1* (log2FC = 11.29) and *ITIH1* (log2FC = 11.20), respectively, were also associated with insulin resistance. *IGFBP1* is a multifunctional protein that regulates the biological effects of insulin-like growth factors, regulates glucose and lipid metabolism, and participates in apoptosis and other biological processes [31]. When insulin levels decrease, *IGFBP1* expression increases, leading to a decrease in the release of glucose; this consequently affects lipid metabolism and promotes fatty acid oxidation [32]. Kim et al. reported that overproduction of *ITIH1* after loss of Gα13 in the liver exacerbates systemic insulin resistance in mice [33].Moreover, the expression levels of *ITIH3* (log_2_FC = 10.89) and *ITIH2* (log_2_FC = 10.79) were significantly upregulated. Glucose and lipid metabolism regulation by controlling insulin sensitivity may be an important transcriptional regulatory mechanism in the intestinal tract of lambs fed high levels of MR.

The most significantly downregulated gene in our study was *MGAT2* (log_2_FC = −11.86). *MGAT2* plays a key role in the regulation of insulin sensitivity and lipid metabolism. It is primarily expressed in the small intestine, liver, and adipose tissue, and its biological effects include catalyzing the synthesis of diacylglycerols from monoacylglycerols in the final step of triglyceride biosynthesis [34]. Further, *MGAT2* reportedly plays a key role in regulating insulin sensitivity as well as delaying triglyceride entry into the blood [35]. Herein the downregulation of *MGAT2* expression seems to promote fatty acid metabolism by inhibiting triglyceride resynthesis in intestinal tissues to maintain a high metabolic level. The inhibition of *MGAT2* can evidently increase free fatty acids in the small intestine epithelium and upregulate the expression of *CYP2C* and *CYP4A*, which are core system members of the cytochrome P450 enzyme superfamily [36]. Interestingly, the expression of several core system members of the P450 enzyme superfamily were also significantly upregulated in this study, particularly *CYP2E1* (log_2_FC = 10.96), which was one of the most significantly upregulated genes. This indicates that there may be a correlation between the expression of *MGAT2* and cytochrome P450 enzyme superfamily genes in intestinal tissues. The primary function of CYP2E1 is to metabolize a wide variety of endogenous and exogenous compounds, including alcohol, drugs, environmental toxins, and carcinogens [37]. In particular, CYP2E1 catalyzes the oxidation of free fatty acids, in addition to other substrates [38]. CYP2E1 evidently plays a role in the metabolism of free fatty acids in the liver, where it contributes to energy production by converting free fatty acids to acetyl-CoA via β-oxidation [39]. In this study, the downregulation of *MGAT2* expression may increase the free fatty acids in the intestinal epithelium by reducing the synthesis of triglycerides, thus increasing the substrate of CYP2E1, which could be the reason for the significant upregulation of *CYP2E1* expression.

The results of WGCNA revealed that among the aforementioned genes with the highest FC, *AHSG*, *IGFBP1*, *ITIH1*, *ITIH2*, *ITIH3*, *CYP2E1*, and *FATUB* all belonged to the same gene module (turquoise), which was significantly positively correlated with intestinal lactase, lipase, and trypsin activities. The gene module containing *MGAT2* (gray) was negatively correlated with the intestinal villus height and digestive enzyme activity. Combined with the biological functions of these genes, our analysis suggested that the insulin sensitivity of lambs in group H was affected via transcriptional regulation under the condition of an increased intestinal epithelial cell renewal rate and metabolic demand, reducing excessive glucose interception by intestinal tissues to ensure adequate glucose release into the portal circulation. At the same time, the reduction of glucose intercepted by the intestinal tissues promotes the degradation of lipids and proteins in intestinal tissues to meet the energy demand of intestinal cells. It is possible to promote the structural and functional development of intestinal tissues by providing adequate levels of lipids or proteins as energy and material sources. This is supported by studies reporting that supplementing lipids or partial replacement of lactose with lipids in MR improves intestinal development of pre-weaned dairy calves [40,41].

In the current study, we found that in the top 20 DEGs with the highest FC, four significantly upregulated genes were associated with the coagulation cascades pathway: *FGA* (log_2_FC = 10.73), *FGB* (log_2_FC = 9.87), *VTN* (log_2_FC = 10.75), and *PLG* (log_2_FC = 11.18). This pathway is also one of the most significantly enriched KEGG pathways in this study. *FGA* and *FGB* are involved in fibrin synthesis [42], and *VTN* inhibits blood clot formation by binding to and regulating the activity of clotting factors and promoting the proliferation and migration of various cell types involved in the repair process [43]. The role of *PLG* is to generate fibrinase under the action of a plasminogen activator, which promotes fibrin breakdown [44]. Fibrin mediates platelet and endothelial cell diffusion, tissue fibroblast proliferation, and capillary angiogenesis [45]. Furthermore, fibrin interacts with other proteins in the extracellular matrix of intestinal tissues, such as fibronectin and vitronectin, to influence cell behavior and tissue remodeling. This interaction affects cell adhesion, migration, and proliferation and contributes to tissue repair and regeneration [43,46]. In this study, upon collectively assessing changes in intestinal metabolism-related genes, intestinal histomorphological indices, and enzyme activity indices, we speculated that the increase in intestinal permeability and metabolic activity of lambs in the high MR group was associated with high requirements for rapid cell renewal, which was the main reason for the upregulation of genes related to fibrin production in intestinal tissues. However, a higher fibrin content may lead to the development of microcirculation thrombus; the upregulation of *PLG* expression may be a feedback regulation. Further studies are warranted to elucidate the mechanism underlying the influence of MR feeding levels on the intestinal complement system.

## 5. Conclusions

Intensive MR feeding promotes intestinal morphological development and digestive enzyme activities, which remain significant even at 14 days postweaning. We observed that intensive MR feeding considerably affected intestinal gene expression in lambs; a total of 1179 DEGs were identified in the jejunum at 14 days postweaning. In addition, intensive MR feeding affects the insulin sensitivity of intestinal tissues and regulates nutrient distribution and metabolism by synchronizing the expression of *AHSG*, *IGFBP1*, *MGAT2*, *ITIH*, and *CYP2E1* in the jejunal tissue of lambs.

## Figures and Tables

**Figure 1 animals-13-01733-f001:**
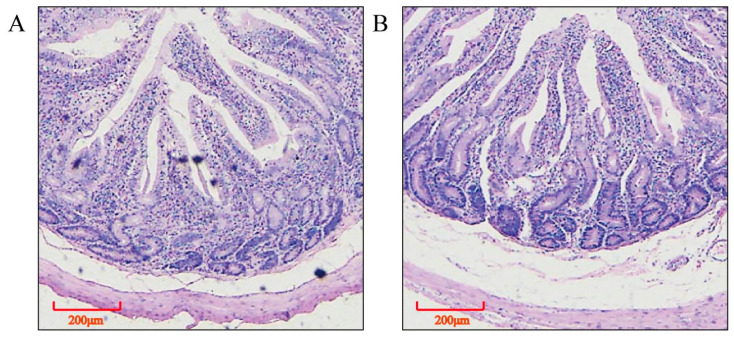
Jejunal histomorphology (hematoxylin eosin staining). (**A**) control group, (**B**) intensive MR feeding group.

**Figure 2 animals-13-01733-f002:**
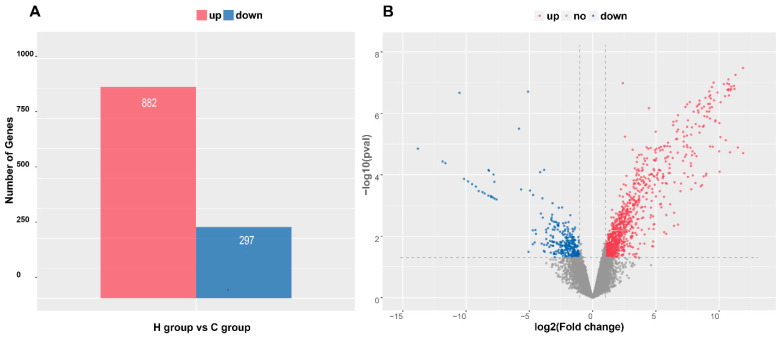
Numbers and distributions of differentially expressed genes (DEGs). (**A**) Numbers of up- and downregulated DEGs. (**B**) Volcano map of DEGs. Note: C: control group, H: intensive MR feeding group.

**Figure 3 animals-13-01733-f003:**
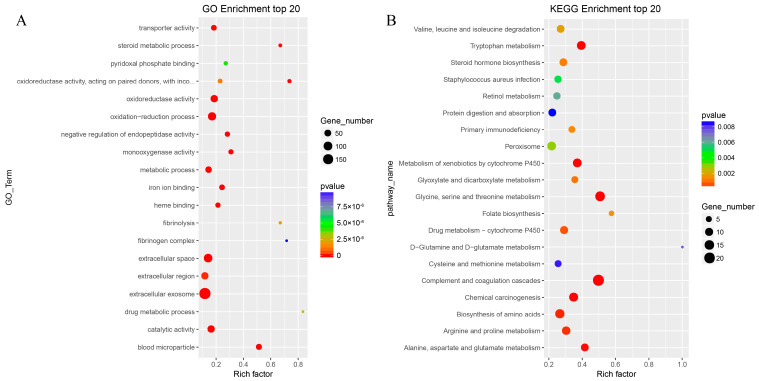
Gene ontology (GO) and Kyoto Encyclopedia of Genes and Genomes pathway (KEGG) enrichment analyses of differentially expressed genes. Top 20 GO (**A**) and KEGG (**B**) enrichments are shown. The X-axis represents enrichment scores, and the Y-axis represents pathway terms. The circle color indicates *p* values, and circle size indicates the number of DEGs. Redder and larger circles indicate that pathway enrichment was higher or the DEG number was larger in the pathway.

**Figure 4 animals-13-01733-f004:**
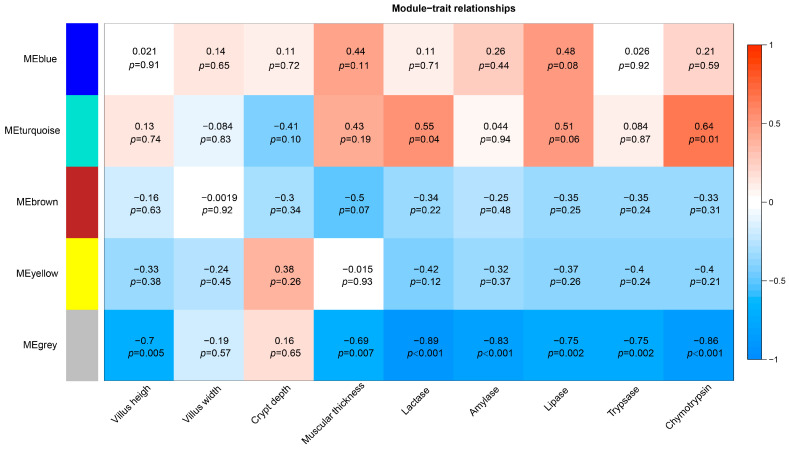
Gene co-expression modules, and correlation coefficients between the eigengene values of these modules and jejunal histomorphological index and digestive enzyme activity.

**Table 1 animals-13-01733-t001:** Ingredients and chemical composition of the starter diet and MR (air-dried basis).

Items	Starter Diet ^1^	MR
Ingredients [%]		
Alfalfa meal	18.50	
Corn	21.00	
Extruded corn	22.30	
Bran	6.00	
Soybean meal	21.50	
Extruded soybean	4.00	
Corn gluten meal	5.00	
Limestone	0.30	
Premix ^2^	1.00	
NaCl	0.40	
Total	100.00	
Chemical composition		
DM (%)	90.96	96.91
DE (MJ·kg^−1^)	13.01	-
CP (%)	19.50	23.22
Fat (%)	1.33	13.20
Starch (%)	33.10	0.00
NDF (%)	18.87	0.00
ADF (%)	8.60	0.00

Notes: ^1^ The starter diet was pelleted. ^2^ The premix included the following per kg of the starter diet: 25 mg Fe as FeSO_4_·H_2_O; 40 mg Zn as ZnSO_4_·H_2_O; 8 mg Cu as CuSO_4_·5H_2_O; 40 mg Mn as MnSO_4_·H_2_O; 0.3 mg I as KI; 0.2 mg Se as Na_2_SeO_3_; 0.1 mg Co as CoCl_2_; 940 IU vitamin A; 111 IU vitamin D; 20 IU vitamin E; and 0.02 mg vitamin B12. MR: milk replacer, DM: dry matter.

**Table 2 animals-13-01733-t002:** Growth performance and feed intake.

Item	Group		SEM	*p* Value
	C	H		
BW (kg)				
7 d	4.64	4.62	0.28	0.969
35 d	8.66	10.43	0.50	0.078
49 d	11.26	12.37	0.60	0.377
ADG (g)				
Preweaning (7–35 d)	143.57	207.43	12.14	0.003
Postweaning (36–49 d)	185.82	139.18	13.97	0.096
Starter intake (g)				
Preweaning (7–35 d)	99.96	58.99	13.06	0.120
Postweaning (36–49 d)	416.40	368.19	28.37	0.417
MR intake (g)				
Preweaning (7–35 d)	111	222	-	-
Postweaning (36–49 d)	-	-	-	-
DM intake (g)				
Preweaning (7–35 d)	201.40	270.51	14.61	0.011
Postweaning (36–49 d)	390.83	345.59	26.63	0.417
FCR (daily DM intake/ADG)				
Preweaning (7–35 d)	1.39	1.32	0.08	0.407
Postweaning (36–49 d)	2.14	2.58	0.20	0.051

Note: C: control group, H: intensive MR feeding group, BW: body weight, ADG: average daily gain, DM: dry matter, FCR: feed conversion ratio, MR: milk replacer.

**Table 3 animals-13-01733-t003:** Jejunal histomorphology and digestive enzyme activity.

Item	Group		SEM	*p* Value
	C	H		
Villus height (μm)	421.96	579.03	28.53	0.001
Villus width (μm)	143.93	153.03	4.29	0.307
Crypt depth (μm)	225.85	211.03	11.85	0.553
Muscular thickness (μm)	128.02	155.10	5.52	0.007
Lactase activity (U/mgprot)	2.80	17.18	2.20	<0.001
Amylase activity (U/mgprot)	0.20	0.49	0.05	<0.001
Lipase activity (U/mgprot)	9.87	38.24	4.96	0.001
Tryptase activity (U/mgprot)	0.19	0.58	0.07	0.002
Chymotrypsin activity (U/mgprot)	0.22	0.86	0.10	<0.001

Note: C: control group, H: intensive MR feeding group.

**Table 4 animals-13-01733-t004:** List of top 20 DEGs with the highest FC.

Gene ID	Gene	Description	Group		log_2_FC	*p* Value	*Q* Value
			C	H			
MSTRG.1830	*AHSG*	alpha-2-HS glycoprotein	0.45	717.51	11.88	<0.001	<0.001
MSTRG.19394	*IGFBP1*	insulin-like growth factor-binding protein 1 precursor	0.18	191.33	11.29	<0.001	<0.001
MSTRG.10283	*ITIH1*	inter-alpha-trypsin inhibitor heavy chain 1	0.07	66.25	11.20	<0.001	<0.001
MSTRG.23484	*PLG*	plasminogen	0.19	188.37	11.18	<0.001	<0.001
MSTRG.14437	*CYP2E1*	cytochrome P450 family 2 subfamily E member 1	0.24	200.13	10.96	<0.001	<0.001
MSTRG.7663	*HPX*	hemopexin	0.30	255.80	10.94	<0.001	<0.001
MSTRG.10280	*ITIH3*	inter-alpha-trypsin inhibitor heavy chain 3	0.22	180.73	10.89	<0.001	<0.001
MSTRG.3079	*CCL16*	C-C motif chemokine ligand 16	0.01	11.80	10.85	<0.001	0.003
MSTRG.5112	*ITIH2*	inter-alpha-trypsin inhibitor heavy chain 2	0.22	164.15	10.79	<0.001	<0.001
MSTRG.3150	*VTN*	vitronectin	0.34	249.51	10.75	<0.001	<0.001
MSTRG.8466	*FGA*	fibrinogen alpha chain	0.35	250.74	10.73	<0.001	<0.001
MSTRG.21609	*ALB*	albumin	3.08	2174.13	10.72	<0.001	<0.001
MSTRG.10688	*AMBP*	alpha-1-microglobulin/bikunin precursor	0.20	138.58	10.64	<0.001	<0.001
MSTRG.14603	*TTR*	transthyretin	0.27	161.47	10.46	<0.001	<0.001
MSTRG.1828	*HRG*	histidine-rich glycoprotein	0.07	41.77	10.37	<0.001	<0.001
MSTRG.21601	*GC*	vitamin D-binding protein	0.75	347.75	10.11	<0.001	<0.001
MSTRG.8468	*FGB*	fibrinogen beta chain	0.51	201.26	9.87	<0.001	<0.001
MSTRG.15211	*ACSM1*	acyl-CoA synthetase medium-chain family member 1	0.13	47.39	9.71	<0.001	0.001
MSTRG.1829	*FETUB*	fetuin-B	0.10	37.77	9.70	<0.001	0.001
MSTRG.22432	*MGAT2*	mannosyl (alpha-1,6-)-glycoprotein beta-1,2-N-acetylglucosaminyltransferase	5.13	0.01	−11.86	<0.001	0.005

Notes: DEGs: differentially expressed genes, FC: fold change, C: control group, H: intensive MR feeding group.

## Data Availability

The sequence files determined in the present study were deposited at the Sequence Read Archive (SRA; http://www.ncbi.nlm.nih.gov/subs/ (accessed on 10 March 2023); SRA accession number: PRJNA938493).

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
