# Peer review of "Transcriptome Analysis to Elucidate the Effects of Milk Replacer Feeding Level on Intestinal Function and Development of Early Lambs"

_animals, 2023, doi:10.3390/ani13111733_

Round 1
Reviewer 1 Report
The manuscript entitled “Transcriptome analysis to elucidate the effects of milk replacer feeding level on intestinal function and development of early lambs” investigated the effects of milk replacer (MR) feeding levels on the intestinal development of lambs. The results showed that higher levels of MR led to increased growth and digestive enzyme activities, as well as changes in gene expression related to nutrient metabolism and energy regulation. It suggests that early MR feeding can impact lamb growth and nutrient absorption. This research seems meaningful and interesting. However, there are some question should be addressed.
1. Why did the author choose the jejunum to perform the RNA-seq analysis.
2. The villus height, villus width and some other index seem to not belong to the fecal microbial richness and diversity indices scope.
3. Why did the author just choose 9 genes to validate the RNA-seq results.
4. There is no 0% MR control group.
Author Response
Point 1 The manuscript entitled “Transcriptome analysis to elucidate the effects of milk replacer feeding level on intestinal function and development of early lambs” investigated the effects of milk replacer (MR) feeding levels on the intestinal development of lambs. The results showed that higher levels of MR led to increased growth and digestive enzyme activities, as well as changes in gene expression related to nutrient metabolism and energy regulation. It suggests that early MR feeding can impact lamb growth and nutrient absorption. This research seems meaningful and interesting. However, there are some question should be addressed.
Author Response: Thank you for reviewing our manuscript and providing valuable feedback and suggestions. We greatly appreciate your time and effort. Based on your comments and suggestions, we have revised the paper accordingly. Please find our point-by-point responses to your comments and suggestions below.
Point 2 Why did the author choose the jejunum to perform the RNA-seq analysis.
Author Response: Thank you for raising this question. We selected the jejunum as the target tissue for RNA-seq analysis due to its relatively greater length in comparison to other intestinal segments in sheep, and its crucial role in facilitating the absorption of nutrients.
Point 3 The villus height, villus width and some other index seem to not belong to the fecal microbial richness and diversity indices scope.
Author Response: I sincerely apologize for providing an incorrect title for Table 3. The mistake was due to a typographical error. We have corrected it. Thanks for catching.
Point 4 Why did the author just choose 9 genes to validate the RNA-seq results.
Author Response: Thank you for raising this question. We selected 12 genes to design primers, but three of them did not meet the amplification efficiency requirements. In RNA-Seq, all transcripts originating from a gene are assigned to that gene. However, most genes have multiple transcripts, but the primers designed for qPCR may not represent all transcripts of the gene. Due to the inherent differences between the two techniques, many transcriptome sequencing studies do not perform qPCR validation. We have included Figure 2 in the supplementary figures for reference.

Reviewer 2 Report
Through studying the intestines of lambs fed with different MR levels, the article found that different MR levels have a certain degree of impact on the development and function of the intestines of lambs, and conducted correlation analysis on the discovered differentially expressed genes and the intestinal traits of lambs, providing certain scientific guidance for the MR feeding of lambs.
Overall, this paper is well organized, but some minor issues need to be improved
1. If there are any figures of Jejunal histomorphology?
2. Please confirm whether the protocol of quantitative reverse-transcription PCR is correct.
3. In material methods, why use a specific adipose type to build an mRNA library?
4. If the numbers in Table 2 of Group refer to FPKM? If yes, please indicate at the end of the table. If not, then what is it?
5. Does the quantitative experiment in Figure 2 conduct a significance test? Please detail.
6. Please verify that the title of result 6 is correct.
7. Please carefully check the article for any grammatical or expressive problems.
Author Response
Point 1 Through studying the intestines of lambs fed with different MR levels, the article found that different MR levels have a certain degree of impact on the development and function of the intestines of lambs, and conducted correlation analysis on the discovered differentially expressed genes and the intestinal traits of lambs, providing certain scientific guidance for the MR feeding of lambs.
Overall, this paper is well organized, but some minor issues need to be improved.
Author Response: Thank you for reviewing our manuscript and providing valuable feedback and suggestions. We greatly appreciate your time and effort. Based on your comments and suggestions, we have revised the paper accordingly. Please find our point-by-point responses to your comments and suggestions below.
Point 2 If there are any figures of Jejunal histomorphology.
Author Response: Thank you for your advice. We have included the figures of jejunal histomorphology in the revised version of the manuscript.
Point 3 Please confirm whether the protocol of quantitative reverse-transcription PCR is correct.
Author Response: Thank you for raising this question. We confirmed that the protocol of qPCR is correct. The method for calculating relative gene expression was added.
Point 4 In material methods, why use a specific adipose type to build an mRNA library?
Author Response: We apologize for the error in our method description. We have rectified it by modifying the sentence as follows: Approximately 10 μg of total RNA was used to isolate poly(A) mRNA with poly(T) oligo-attached magnetic beads (Invitrogen, CA, USA).
Point 5 If the numbers in Table 2 of Group refer to FPKM? If yes, please indicate at the end of the table. If not, then what is it?
Author Response: Thank you for raising this question. I think you may refer to Figure 2. The relative gene expression was calculated as 2−ΔΔCT, and the fold change was visualized in the figure. We have added relevant explanations at the end of the figure.
Point 6 Does the quantitative experiment in Figure 2 conduct a significance test? Please detail.
Author Response: Thank you for raising this question. Most genes have multiple transcripts, and some have complex transcript forms. In RNA-Seq, all transcripts originating from a gene are assigned to that gene. However, the primers designed for qPCR may not represent all transcripts of the gene. Thus, the purpose of using qPCR to validate RNA-Seq in this experiment is to determine the consistency of gene expression trends without conducting differential significance testing. Due to the inherent differences between the two techniques, many transcriptome sequencing studies do not perform qPCR validation. We have included Figure 2 in the supplementary figures for reference.
Point 7 Please verify that the title of result 6 is correct.
Author Response: I apologize sincerely for providing an incorrect title for Result 6. The error was due to a typographical mistake, which we have now rectified. Thanks for catching.
Point 8 Please carefully check the article for any grammatical or expressive problems.
Author Response: Thank you for your valuable feedback. We have carefully reviewed the article to address any potential grammatical or expressive issues. We have also enlisted the help of a professional editor to ensure the quality of the writing in the manuscript.

Reviewer 3 Report
· The purpose of this project was to investigate the effects of different amounts of milk replacer in lamb diets on growth, intestinal development and the intestinal transcriptome. This project was designed well and is of importance for producers and feed manufacturers. Overall, the paper and the data are presented in a clear logical manner that is easy to follow. However, a few clarifications and changes are needed.
o After day 3, when the lambs were separated from the ewes, were the lambs still housed indoors. This needs clarified in the manuscript.
o In table 2, if we have dry matter intake and weight gain, gain to feed ratios are needed. This will need to be updated in table 2, the methods, results, and discussion.
o Preweaning ADG was greater in the H group than the C lambs, however by day 49 there was no difference in the weights of lambs between the two treatment groups. Do you contribute this to weaning stress, as you suggested in the discussion, or what else do you think is going on. This will need to be updated in the discussion.
· Line by line comments:
o Line 104: change our to the
o Line 195: Do not start sentences with abbreviations. Fix here and throughout.
o Line 196: Italicize p-values. Fix here and throughout
o Line 277: change higher to increased
Author Response
Point 1 The purpose of this project was to investigate the effects of different amounts of milk replacer in lamb diets on growth, intestinal development and the intestinal transcriptome. This project was designed well and is of importance for producers and feed manufacturers. Overall, the paper and the data are presented in a clear logical manner that is easy to follow. However, a few clarifications and changes are needed.
Author Response: Thank you for reviewing our manuscript and providing valuable feedback and suggestions. We greatly appreciate your time and effort. Based on your comments and suggestions, we have revised the paper accordingly. Please find our point-by-point responses to your comments and suggestions below.
Point 2 After day 3, when the lambs were separated from the ewes, were the lambs still housed indoors. This needs clarified in the manuscript.
Author Response: We apologize for the lack of detail in the methodology described in our manuscript. From day 4 to day 6, the lambs were trained to use the nipple bottle containing MR, while still being raised with ewes and receiving breast milk. At 7 days of age, lambs were separated from ewes and housed in individual pens; lactation was completely replaced by artificial feeding with MR. We have made revisions and added pertinent information.
Point 3 In table 2, if we have dry matter intake and weight gain, gain to feed ratios are needed. This will need to be updated in table 2, the methods, results, and discussion.
Author Response: Thank you very much for your suggestion. We have incorporated the data on feed conversion ratio into the manuscript, and have also made revisions and added relevant information to the corresponding sections.
Point 4 Preweaning ADG was greater in the H group than the C lambs, however by day 49 there was no difference in the weights of lambs between the two treatment groups. Do you contribute this to weaning stress, as you suggested in the discussion, or what else do you think is going on. This will need to be updated in the discussion.
Author Response: Thanks for your suggestion. We have revised the discussion section of the paper based on your feedback, and have incorporated the possible causes of the observed results from the perspective of weaning stress and rumen development, supported by relevant literature. The H group exhibited increased preweaning ADG compared to the C group, yet there was no significant difference in the lambs' BW between the two treatment groups by day 49. Although there was no significant difference in DM intake after weaning, lambs in group H tended to exhibit lower ADG and higher FCR than those in group C, which may be due to the fact that lambs in group H were more susceptible to weaning stress. Studies have shown that weaning stress can be recovered in the short term, and adverse effects do not usually last long. Additionally, research has indicated that intensive MR feeding have detrimental effects on rumen development, which could be a contributing factor to the temporary decline in growth performance observed in lambs after weaning.
Point 5 Line by line comments:
o Line 104: change our to the
o Line 195: Do not start sentences with abbreviations. Fix here and throughout.
o Line 196: Italicize p-values. Fix here and throughout
o Line 277: change higher to increased
Author Response: Thank you very much for your careful review. We have made revisions according to your suggestions, and have checked and modified similar issues in the whole manuscript.

Round 2
Reviewer 1 Report
The author should explain that why there is no 0% MR control group in last comment.
Author Response
We apologize for the oversight in our previous response to your comment. We have found that we did indeed prepare a response to your comment, but unfortunately, it was accidentally left out when we submitted our revised manuscript. We deeply regret this error and assure you that it was not intentional.
We understand the importance of addressing all reviewer comments thoroughly and regret any inconvenience this may have caused. Please know that we take your feedback seriously and have taken steps to ensure that all comments are addressed and incorporated into our revised manuscript. The following is the response to this review comment:
Point 5 There is no 0% MR control group. Author Response: From a trial design perspective, setting up a 0% MR Group would have been ideal. However, in this experiment, individual housing was implemented for each lamb to attain accurate measurement of feed intake and minimize inter-individual variation. Consequently, since relying solely on solid feed intake may not ensure the survival of 7-day-old lambs adequately, the provision of MR was deemed necessary in the absence of ewes.
Again, we apologize for the oversight, and we appreciate your understanding and continued support in the review process.
Round 3
Reviewer 1 Report
It can be acceted.
Author Response
We would like to sincerely thank you for the time and effort you dedicated to reviewing our manuscript and for agreeing to accept our paper.